# Mass Spectrometry-Based Characterization of the Virion Proteome, Phosphoproteome, and Associated Kinase Activity of Human Cytomegalovirus

**DOI:** 10.3390/microorganisms8060820

**Published:** 2020-05-30

**Authors:** Yohann Couté, Alexandra Kraut, Christine Zimmermann, Nicole Büscher, Anne-Marie Hesse, Christophe Bruley, Marco De Andrea, Christina Wangen, Friedrich Hahn, Manfred Marschall, Bodo Plachter

**Affiliations:** 1University Grenoble Alpes, CEA, Inserm, BIG-BGE, 38000 Grenoble, France; alexandra.kraut@cea.fr (A.K.); anne-marie.hesse@cea.fr (A.-M.H.); christophe.bruley@cea.fr (C.B.); 2Institute for Virology and Forschungszentrum für Immuntherapie, University Medical Center of the Johannes Gutenberg-University Mainz, Obere Zahlbacher Str. 67, D-55131 Mainz, Germany; chrzimme@uni-mainz.de (C.Z.); bueschni@uni-mainz.de (N.B.); 3Department of Public Health and Pediatric Sciences, Turin Medical School, University of Turin, 10126 Turin, and CAAD – Center for Translational Research on Autoimmune and Allergic Disease, Novara Medical School, 28100 Novara, Italy; marco.deandrea@unito.it; 4Institute for Clinical and Molecular Virology, Friedrich-Alexander University of Erlangen-Nürnberg (FAU), 91054 Erlangen, Germany; christina.wangen@uk-erlangen.de (C.W.); friedrich.hahn@uk-erlangen.de (F.H.); manfred.marschall@fau.de (M.M.)

**Keywords:** human cytomegalovirus, virion composition, proteins, phosphorylation, virion-associated kinase, mass spectrometry-based proteomics

## Abstract

The assembly of human cytomegalovirus (HCMV) virions is an orchestrated process that requires, as an essential prerequisite, the complex crosstalk between viral structural proteins. Currently, however, the mechanisms governing the successive steps in the constitution of virion protein complexes remain elusive. Protein phosphorylation is a key regulator determining the sequential changes in the conformation, binding, dynamics, and stability of proteins in the course of multiprotein assembly. In this review, we present a comprehensive map of the HCMV virion proteome, including a refined view on the virion phosphoproteome, based on previous publications supplemented by new results. Thus, a novel dataset of viral and cellular proteins contained in HCMV virions is generated, providing a basis for future analyses of individual phosphorylation steps and sites involved in the orchestrated assembly of HCMV virion-specific multiprotein complexes. Finally, we present the current knowledge on the activity of pUL97, the HCMV-encoded and virion-associated kinase, in phosphorylating viral and host proteins.

## 1. Introduction

The human cytomegalovirus (HCMV) is a member of the *Betaherpesvirinae* subfamily. Seroprevalence studies have indicated its wide presence in the population worldwide [1]. While HCMV infection is mainly asymptomatic in immunocompetent hosts, active replication, either from primary infection, reinfection, or reactivation from latency, causes life-threatening complications in immunocompromised patients, as well as serious fetal damage during pregnancy [2,3,4,5]. HCMV infects and replicates in a broad range of cell types, notably thanks to the remarkable modularity of protein complexes present in its envelope [6]. Its complex infectious cycle involves the hijacking of numerous host pathways, notably through the modulation of various functional protein complexes [7]. This culminates with the production of precisely packaged virions composed of four main structural components: the core genome contained within the capsid, which itself is coated in a protein matrix called the tegument, connected to a surrounding outer lipid bilayer envelope [8].

Virion assembly is a multistep process that takes place sequentially in different structures within infected cells. In the nuclei of host cells, the viral genome is replicated and the capsids are assembled before the genomes are encapsidated [9]. These immature-stage capsids are already decorated with an internal layer of tegument proteins before their budding through the nuclear membranes [10]. Nucleocytoplasmic transport of capsids, termed nuclear egress, is mediated through a multistep regulatory process involving the reorganization of the nuclear lamina and the primary envelopment and subsequent de-envelopment of nuclear capsids, in which the virus-encoded protein kinase plays a central role through the phosphorylation of various host and viral proteins [11]. Maturation of virions then continues within the virus-induced cytoplasmic assembly complex [12,13], which contains Golgi-derived structures, as well as early and recycling endosome-derived structures [14]. In the assembly complex, most of the tegument proteins become associated and ultimately virions are secondarily enveloped in their mature form. In the final step, virions are transported to the plasma membrane within vesicles and released in the extracellular space through fusion of the vesicle and plasma membranes [8,15]. Thus, as a key feature, the morphogenesis of HCMV virions requires the formation of multiprotein complexes [7,16,17].

Mature HCMV particles show a remarkably high level of complexity and conservation with respect to their protein composition, independent of the viral strains that were analyzed [18,19,20]. This is paralleled by conserved patterns of viral protein levels in infected cells independent of HCMV strains [21]. The underlying regulatory mechanisms that govern HCMV virion assembly in such a reproducible fashion in infected cells are, however, still widely unknown. One well-established key regulatory switch in protein interaction is phosphorylation. Although the molecular events that drive phosphorylation-dependent protein interactions are well studied, e.g., in signal transduction, there is only limited information on how herpesvirus particle assembly may be modulated by the phosphorylation status of individual virion components. Indeed, this process requires sequential steps of protein interactions, resembling physiological processes in infected cells, leading to the formation of multiprotein complexes. Phosphorylation could, thus, be considered as a key regulatory switch in herpesvirus assembly. It is known that some of the HCMV virion proteins, particularly in the tegument, are phosphorylated [22,23]. The same proteins have also been found to be phosphorylated during their intracellular state of infection [24,25,26,27,28]. Here, an interesting question addresses the issue how phosphorylation and specific patterns of phosphosite usage relate to virion assembly and morphogenesis.

During the past twenty years, the constant evolution of mass spectrometry (MS)-based proteomics has revolutionized the throughput, depth, and precision of protein characterization in many contexts [29], including HCMV molecular biology [30]. Here, we highlighted how MS-based proteomics has been central to characterizing the protein content of HCMV virions, identifying viral and cellular players. Furthermore, we combined data from the literature with a novel dataset we produced, aiming to provide a comprehensive map of the protein composition of HCMV infectious particles. We then integrated information concerning the phosphorylation status of proteins within virions, combining published and novel datasets. Furthermore, a side-by-side analysis of the phosphoproteomes of HCMV virions and infected cells revealed marked differences in the phosphorylation patterns of viral proteins. This refined information may provide a basis for future analyses of specific viral phosphosite occupancy involved in the regulation of the orchestrated assembly of HCMV virions. Finally, we reviewed the current knowledge on the phosphorylation activity of the HCMV-encoded and virion-associated kinase pUL97, while providing original results on its phosphorylation of the host protein IFI16.

## 2. Materials and Methods for Original Datasets 

### 2.1. Purification of HCMV Virion

For virion purification, 1.8 × 10^6^ primary human foreskin fibroblasts (HFF) cells (passage numbers 17 to 19) were grown in 175 cm^2^ tissue culture flasks (Greiner, Frickenhausen, Germany) in minimal essential medium (MEM; Gibco-BRL via Thermo Scientific, Waltham, MA, USA) supplemented with 5% FCS (Biochrom, Berlin, Germany), L-glutamine (2mM, Sigma-Aldrich CO, St. Louis, MO, USA), and gentamicin (50 mg/L, Gibco-BRL via Thermo Scientific, Waltham, MA, USA) for 1 day. The cells were infected with 0.5 mL of a frozen stock of the HCMV strain AD169-RV [31] (passage number 6) at a rate of infection of 0.6 in 3.5 mL of MEM. The virus inoculum was allowed to adsorb for 1.5 h at 37 °C. Subsequently, the culture medium was supplemented and the cells were incubated for at least 7 days. When the cells showed a cytopathic effect (CPE) of late HCMV infection (usually at day 7 post-infection (p.i.)), the supernatant was harvested and centrifuged for 10 min at 2800 rpm using a TX-400 rotor in a Heraeus Megafuge 16 (Thermo Scientific, Waltham, MA, USA) to remove cellular debris. Virion preparation was then performed according to a modification of the procedure suggested by Irmiere and Gibson [32]. The supernatant was collected and centrifuged at 95,000× *g* (70 min; 10 °C) in a SW45Ti rotor in a Beckman Optima L-90K ultracentrifuge (Beckman Coulter, Brea, CA, USA). The pellets were re-suspended in 700 µl of 1× phosphate-buffered saline (PBS, Sigma-Aldrich CO, St. Louis, MO, USA). Glycerol tartrate gradients were prepared immediately before use. For this, 4 mL of a 35% Na-tartrate solution (Roth, Karlsruhe, Germany) in 0.04 M Na-phosphate (Roth, Karlsruhe, Germany) buffer at pH 7.4 was applied to one column, while 5 mL of a 15% Na-tartrate–30% glycerol (Roth, Karlsruhe, Germany) solution in 0.04 M Na-phosphate buffer at pH 7.4 was applied to the second column of a gradient mixer. The gradients were prepared by slowly dropping the solutions into Beckman Ultraclear centrifuge tubes (14 by 89 mm, Beckman Coulter, Brea, CA, USA), positioned at an angle of 45°. The solution containing viral particles was then carefully layered on top of the gradients. Ultracentrifugation was performed with breaks and minimum deceleration in a Beckman SW41 swing-out rotor (Beckman Coulter, Brea, CA, USA) for 60 min at 90,000× *g* and 10 °C. The particles were illuminated by light scattering and collected from the gradient by penetrating the centrifuge tube with a hollow needle below the band. Samples were carefully drawn from the tube with a syringe. The particles were washed with 1× PBS and pelleted in an SW41 swing-out rotor (Beckman Coulter, Brea, CA, USA) for 90 min at 98,000× *g* and 10 °C. After the last centrifugation step, the virions were resuspended in 100 µl of 2× Laemmli buffer [33], centrifuged at 9390× *g* in a FA-45-24-11 rotor in an Eppendorf 5424R centrifuge (Eppendorf, Hamburg, Germany), and stored at −80 °C. The protein composition of purified virions was analyzed by SDS-PAGE. For this, extracted proteins were loaded on a 10% Bis-Tris-polyacrylamide gel (Invitrogen by Thermo Fisher Scientific, Waltham, MA, USA). After electrophoresis, the gel was incubated in 20 mL Instant Blue staining solution for 30 min according to the manufacturer’s protocol (Expedeon via Sigma Aldrich CO, St. Louis, MO, USA). 

### 2.2. MS-Based Proteomic Analyses of Purified Virions

Similar amount of extracted proteins from the three biological replicates were stacked in the top of a SDS-PAGE gel (NuPAGE Novex 4-12% Bis-Tris, Gels, 1.0 mm, Life Technologies SAS, Courtaboeuf, France) and in-gel digested with trypsin (Promega Corporation, Madison, WI, USA) as previously described [34]. Resulting peptides were either directly analyzed by nano-liquid chromatography (nanoLC)-MS/MS analyses or submitted to phosphopeptide enrichment using titanium dioxide beads (TitanSphere, GL Sciences, Inc. via Interchim, Montluçon, France) as previously described [35].

NanoLC-MS/MS analyses were performed using an Ultimate 3000 RSLCnano coupled to a Q-Exactive Plus (Thermo Fisher Scientific, Bremen, Germany). Peptides were sampled on a 300 μm × 5 mm PepMap C18 precolumn (Thermo Scientific, Waltham, MA, USA) and separated on a 75 μm × 250 mm C18 column (Reprosil-Pur 120 C18-AQ, 1.9 μm, Dr. Maisch GmbH, Ammerbuch, Germany). The nanoLC method consisted of a 120 min gradient at a flow rate of 300 nl/min, ranging from 5.1% to 41% acetonitrile (Sigma-Aldrich CO, St. Louis, MO, USA) in 0.1% formic acid (Merck KGaA, Darmstadt, Germany). The spray voltage was set at 1.5 kV and the heated capillary was adjusted to 250 °C. Survey full-scan MS spectra (m/z = 400–1600) were acquired at a resolution of 70,000, with automatic gain control (AGC) set to 10^6^ ions (maximum filling time 250 ms) and with the lock mass option activated. The 10 most intense ions were fragmented by higher-energy collisional dissociation (normalized collision energy = 30) at a resolution of 17,500, with AGC set to 10^6^ ions (maximum filling time of 250 ms and minimum AGC target of 3 × 10^3^) and dynamic exclusion set to 20 s. MS and MS/MS data were acquired using the Xcalibur software (Thermo Fisher Scientific, Bremen, Germany).

Peptides and proteins were identified using Mascot [36] v. 2.6.0 (Matrix Science, London, UK) through concomitant searches in the Uniprot database (*Homo sapiens* and HCMV AD169 taxonomies, October 2019 version), a classical contaminant database, and the corresponding reversed databases. Trypsin/P was chosen as the enzyme and two missed cleavages were allowed. Precursor and fragment mass error tolerances were set at 10 and 25 ppm, respectively. Peptide modifications allowed during the search were: carbamidomethyl (C, fixed), acetyl (protein N-term, variable), oxidation (M, variable), and phosphoration (STY, variable). The Proline software [37] was used to filter the results according to conservation of rank 1 peptides, peptide-spectrum-match score ≥25, peptide length ≥6 amino acids, false discovery rate of peptide-spectrum-match identifications <1% as calculated on peptide-spectrum-match scores by employing the reverse database strategy, and minimum of 1 specific peptide per identified protein group. Proline was then used to perform compilation, grouping, and comparison of the protein groups and phosphosites identified in the different samples. Proteins from the contaminant database and additional keratins were discarded from the final list of identified proteins. Only phosphopeptides with a localization probability above 75%, as calculated using the Mascot Delta Score [38], were conserved.

Raw and processed MS data were deposited to the ProteomeXchange Consortium via the PRIDE (PRoteomics IDEntifications database) partner repository [39] with the dataset identifier PXD012921.

## 3. MS-Based Characterization of the HCMV Virion Proteome

Different strategies may be implemented to characterize the protein content of a biological system. The most widely used approach is based on an enzymatic digestion of the extracted proteins before analysis of the resulting peptides by liquid chromatography (LC) coupled to tandem MS using soft ionization. Dedicated computational tools are then used to identify the peptides and proteins, and if needed to extract their abundances. Importantly, the MS instruments and specific algorithms are constantly improved to make proteomic analyses more powerful [29,40]. Soon after the pioneering work of Pappin et al. reporting that masses and fragmentation patterns of proteolytic peptides can be used to rapidly identify proteins from databases [41], this strategy was applied to characterize the proteome of a virion for the first time, namely Ictalurid herpesvirus 1 (also known as Channel Catfish Virus), a member of the Alloherpesviridae. This resulted in the identification of 12 proteins—11 expressed from the viral genome and 1 from the host genome [42]. Following this, the proteomes of various viruses have been investigated using MS-based proteomics [43,44,45,46].

Seminal works aiming at characterizing the protein content of HCMV virions have concluded that about 30 different proteins were packaged, with some of them being phosphorylated [32,47,48,49,50]. The application of advanced MS-based proteomic approaches demonstrated that the proteome content of HCMV virions was much more complex than expected. To date, three different datasets have been delivered that have characterized the proteome of the AD169 laboratory strain of HCMV or of AD169-derived clones [18,19,23]. In 2004, Varnum et al. performed the first characterization of the HCMV virion proteome. For this, they purified infectious particles from the cell culture supernatant of HCMV-infected human dermal fibroblasts before lysis in urea and protein digestion using trypsin. Resulting peptides were fractionated by strong cation exchange and analyzed by reversed-phase nanoLC coupled to tandem MS using an ion trap and a Fourier-transform ion cyclotron resonance (FTICR)-MS for high-accuracy mass measurements. This strategy allowed the identification of 59 proteins encoded by the viral genome using SEQUEST [51] and ICR-2LS [52]. In 2014, Reyda et al., while analyzing the role of pp65 in virion morphogenesis, delivered a complementary proteome of HCMV infectious particles. They purified AD169-RV (also called RV-HB15) virions from the cell culture supernatant of HCMV-infected human foreskin fibroblasts and lysed them in a buffer containing urea, thiourea, and CHAPS (3-[(3-cholamidopropyl)dimethylammonio]-1-propanesulfonate). The extracted proteins were then digested with trypsin using a modified filter-aided sample preparation method [53,54]. Resulting peptides were analyzed by reversed-phase nanoLC coupled to tandem MS using a high-resolution hybrid quadrupole time-of-flight, leading to the identification of 51 viral proteins with at least 2 peptides using the ProteinLynx Global SERVER (Waters). The proteome of HCMV virions of the AD169 strain was further investigated by Rieder et al. [23]. After purification, virions were lysed in urea before in-solution digestion of proteins using trypsin or chymotrypsin. Obtained peptides were analyzed by reversed-phase nanoLC coupled to tandem MS using a high-resolution hybrid quadrupole orbitrap, before identification using the MaxQuant software [55]. This work ended up with the identification of 62 viral proteins. Finally, we provide here a fourth dataset aiming at characterizing the HCMV virion proteome. For this, we prepared three biological replicates, corresponding to three independent cultures, of HCMV virions as described in [19]. This time, particle lysis was performed directly in Laemmli buffer. A control of the independent preparations using SDS-PAGE and Instant Blue staining showed that their protein patterns were very similar, demonstrating the reproducibility of the purification process (Figure 1). For MS-based proteomic analyses, the proteins were stacked in a single band in the top of a SDS-PAGE gel and digested in-gel using trypsin. Extracted peptides were submitted to reversed-phase nanoLC coupled to tandem MS using a high-resolution hybrid quadrupole orbitrap. Data were processed using Mascot [36] and Proline [37], allowing identification of 73 viral proteins (67 in the three replicates, and three each in two and one replicates), representing the highest number of viral proteins identified in purified HCMV virions.

In total, 82 different viral proteins were identified in purified HCMV virions by these four studies using MS-based proteomics (Figure 2, Appendix A). These proteins can be grouped according to their localization within virions [8,56]. In addition to the five constituents of the HCMV capsid (major capsid protein/MCP, pUL46/Trx1, pUL85/Trx2, pUL48a/SCP, and pUL104/PORT), as well as the three capsid-associated proteins pUL93/CVC1, pUL77/CVC2, and pUL80.5, these works revealed the presence of 12 envelope proteins, 24 tegument proteins, and 38 proteins with unknown localization within the viral particle (Appendix A). The majority of the 82 proteins (i.e., 46 proteins) were identified in the 4 studies, whereas 7, 11, and 18 proteins were identified in three, two, and one of the investigations, respectively (Figure 2). This may be explained by differences in the experimental settings of these studies, notably the virion lysis strategies employed, but also the mass spectrometers and data processing tools used, as well as the completeness of the database employed. On that latter point, it must be emphasized that the number of different proteins potentially expressed from the HCMV genome is higher than initially expected [57]. One protein that was reproducibly detected in our novel dataset but not in others is pUL128. This protein is a component of a pentameric envelope protein complex, consisting of glycoproteins H and L, as well as pUL130 and pUL131 [58]. The pentameric complex (PC) has been shown to be important for HCMV infection of cell types such as endothelial cells and epithelial cells [58,59,60,61]. Laboratory HCMV strains are devoid of the PC because of mutations in the UL128-131 gene region. The AD169 strains used for MS-based analyses are devoid of the expression of pUL131, but express UL128 in infected cells [62]. It was, however, surprising that pUL128 was found in the virion proteome, as PC-formation was thought to be essential for packaging of pUL128 into virions. One explanation for this would be that pUL128 was fortuitously packaged into virions. Alternatively, the pUL128 might be packaged in low amounts into virions independent of PC formation. Interestingly, seven proteins with a localization still unknown within HCMV virions have been reproducibly identified in the different studies: two are G-protein-coupled receptor homologues (pUL33 and pUS27), three are known to be involved in DNA replication (pUL44, pUL84, and pUL112/113), one plays a role in transcription (pUL122), and the last one is a member of the nuclear egress complex (pUL50). It might, thus, be hypothesized that these proteins are purposely packaged and that their presence in the virion serves a function during the initial stages of viral infection.

By taking advantages of the possibility to estimate absolute quantities of proteins from label-free MS-based proteomic analyses [63,64], these works allowed the rough estimation of the stoichiometry of proteins identified within HCMV virions. For this purpose, Varnum et al. averaged the extracted intensities of the most abundant peptides identified for each protein, while Reyda et al. used the TOP3 strategy, in which the abundance of a protein is calculated as the average intensity of its three best ionizing peptide. For our original dataset we implemented the intensity-based absolute quantification (iBAQ) approach, in which the intensities of the different peptides identified for a protein are summed before normalization by the number of theoretically observable peptides for each protein [65]. The estimated relative abundances between virion proteins were consistent for many of them across the different studies (Appendix A, extracted abundances normalized to MCP value). For some proteins, however, discrepancies were observed that may be explained by the different experimental strategies used. Importantly, the stoichiometries inferred by label-free MS-based proteomics were close to those obtained through high-resolution characterization by cryo-electron microscopy of the structure of the tegument-coated HCMV capsid [66]. The only exception was for pUL48a/SCP, for which copy numbers were overestimated by Varnum et al. and underestimated by Reyda et al. and in our original dataset as compared to the atomic structure derived from cryo-electron microscopy analysis, showing that each copy of this protein binds exactly one copy of the MCP [49] (Appendix A). Globally, the different MS-based proteomic studies agree on the fact that the tegument proteins represent the main fraction of the protein amount within HCMV virions (55–72%), with pp65 always being estimated as the most abundant resident protein, followed by capsid proteins (16–32%), envelope proteins (11% to 20%), and proteins of unknown localization (1–3%) (Appendix A). This remarkable reproducibility between studies argues in favor of a highly conserved process of HCMV virion morphogenesis, and notably for tegument assembly despite its amorphous nature in ultrastructural analyses. This is underlined by our recent findings that the virion proteome is conserved between different HCMV strains, including primary clinical isolates, and the TB40-BAC4 strain, which expresses the pentameric complex [20]. This is also consistent with previous studies showing that comparable relative amounts of tegument proteins are synthesized in infected cells, irrespective of the HCMV strain used for infection [21].

A common feature of most of the MS-based analyses of virion proteomes is the identification of host proteins [43,44,45,46]. This is also the case for HCMV virions, for which an increasing number of human proteins have been identified as the analyses were carried out using increasingly sensitive MS instruments. Indeed, Varnum et al. reported the identification of 71 host proteins, while the dataset of Reyda et al. included more than 400 different human proteins, the work of Rieder et al. identified more than 1300 human proteins, and our novel dataset contains over 1500 host proteins. The estimation of their abundance within virions suggest that some of them might be rather abundant in virions (Appendix A, extracted abundances normalized to MCP value). However, importantly, the abundance estimates of host proteins identified in the different studies are much less consistent than those observed for viral proteins. Although ultracentrifugation allows for the separation of HCMV virions from cellular debris, contamination of the virion material by cellular components with similar sedimentation properties or unspecific attachment of abundant cellular proteins to the virions cannot be ruled out. Consequently, all proteomic data referring to the packaging of cellular proteins into HCMV virions have to be considered with care. However, several host proteins may be selectively incorporated within virions to play specific roles that remain to be elucidated. Interestingly, numerous proteins were reproducibly identified in the different MS-based studies (63 proteins are found in all datasets and almost 1000 are found in at least 2 datasets). Among them is Annexin A2, one of the most abundant host proteins in our accompanying dataset, which was already described to be present on the surface of virions, playing an important role in HCMV infectivity [67,68]. The question whether other host proteins present on the virion surfaces may be important for viral entry mechanisms or overall infectivity of viral particles still has to be addressed in greater details and can only be speculative at this stage. An immunological functionality of surface-exposed host proteins, in particular glycoproteins, seems plausible, but requires further experimental evidence. Other proteins, such as the type 1 and 2A serine–threonine protein phosphatases, have been shown to be packaged within virions and serve important functions during HCMV infection [69,70]. The biological significance of the presence of other host proteins in mature virions and their location within particles still need to be explored.

## 4. MS-Based Characterization of the HCMV Virion Phosphoproteome

Post-translational modifications (PTMs) of proteins, mainly involving the enzymatic addition of chemical moieties to specific amino acids, are known to play a major role in regulating their localization, turnover, activity, or interaction with other molecules. To date, several hundreds of PTMs have been reported and MS-based proteomics has become the method of choice to identify and quantify modified proteins and sites, in particular thanks to the development of specific sample preparation procedures [71]. Phosphorylation is a widespread PTM that has been extensively explored by MS-based proteomics using complementary approaches [72]. It was notably shown to decorate three-quarters of the detected proteins in human cells to various degrees [73]. A major role played by phosphorylation is intervening in the stabilization or destabilization of protein–protein interactions [74,75,76,77,78,79]. Therefore, the knowledge on the phosphorylation status of HCMV virion proteins may help in understanding how this particular PTM influences the assembly and stability of this multiprotein system.

Although the presence of phosphorylated proteins in HCMV virions was described in the early 1980s [22,80,81,82], it was only recently that their phosphoproteome was investigated on a global scale using a MS-based proteomic approach. For this, Rieder et al. first enriched phosphopeptides by metal oxide affinity chromatography using titanium dioxide beads before analysis by nanoLC-MS/MS [23]. This led to the identification of 83 different phosphosites with localization probability above 75% and belonging to 22 viral proteins (of note, one phosphosite was common to pUL122 and pUL123). Here, we completed this dataset using three biological replicates. For this, we also used titanium dioxide beads to enrich phosphopeptides from in-gel digests of virion proteins. After nanoLC-MS/MS analyses, we identified 142 phosphosites with localization probabilities above 75% and present on 34 different proteins (98 phosphosites were identified in the three biological replicates, while 22 were identified in two and one replicates; Appendix A). In total, these two independent and complementary studies allowed confident identification of 168 phosphosites (137 serine, 27 threonine, and 4 tyrosine residues) on 37 different proteins, demonstrating the unexpected complexity of the HCMV virion phosphoproteome (Figure 3, Appendix A). The differences observed between the two datasets may be explained by the different experimental strategies used, notably virion lysis (urea-containing buffer versus sodium dodecyl sulfate-containing Laemmli buffer), the protein digestion step (in-solution trypsin and chymotrypsin digestions versus in-gel trypsin digestion), and computational tools (MaxQuant versus Mascot and Proline). As expected, a large majority of these phosphosites was found to be present on 15 tegument proteins (122 out of 168), while 12 were identified on 5 capsid and capsid-associated proteins, 16 on 5 envelope proteins, and 18 on 12 proteins with unknown localization in virions (Appendix A). Some viral proteins were found to be extensively phosphorylated, such as the tegument proteins pUL32/pp150, pUL83/pp65, pUL69, and pUL25/pp85 (Figure 4a–f).

During the infectious cycle, phosphorylation is involved in the regulation of many crucial steps, targeting viral and host proteins [30,88]. Interestingly, the comparison of the phosphosites identified on virion proteins with those mapped by a previous investigation aiming to characterize the infected cell phosphoproteome revealed differences for many viral proteins (Figure 3, Appendix A). Striking examples are the viral kinase pUL97 and the viral transactivator pUL82/pp71 (Figure 4e,f). It might be hypothesized that phosphosite occupancy on specific proteins is tightly regulated to allow their orchestrated interaction in the course of virion assembly. This would make sense knowing that the viral kinase pUL97 and the phosphatases PP1-alpha and 2A are packaged within virions [69,70,89,90]. However, although both pUL97 and phosphatases have been shown to be required for critical cellular processes during HCMV infection [88,91,92], it remains unclear if they are involved in regulating the phosphorylation state of individual proteins during virion assembly. The availability of a comprehensive map of phosphosites decorating virion proteins will now allow detailed analyses of the role played by individual phosphosites in the proper assembly of HCMV infectious particles. 

In addition to phosphosites identified on viral proteins, these global phosphoproteomic studies revealed 201 phosphosites present on host proteins, with some of these proteins being reproducibly identified in purified HCMV virions, such as the catalytic subunits of serine–threonine protein phosphatase PP1-alpha and beta (Appendix A). However, only 26 phosphosites observed on host proteins were reproducibly identified in Rieder et al. [23] and in our novel dataset, meaning any conclusions have to be made cautiously.

The functional relevance of the phosphosites decorating viral and host proteins in HCMV virions has not been sufficiently addressed to date. Further experiments should explore the involvement of some of them in the correct assembly of the infectious particles, but also in viral entry, virion infectivity, or other mechanisms. Given the possibility that at least some of these phosphosites are necessary for viral replication, in particular for virion morphogenesis, they might be susceptible to the antiviral potential of HCMV-specific kinase inhibitors (such as the pharmacological inhibitor maribavir, directed against the viral kinase pUL97, which is presently investigated in phase 3 clinical trials). The development of novel types of antiviral drugs targeting specific virus-supportive phosphosites may be considered promising for translational use.

## 5. HCMV Virion-Associated pUL97 Kinase: Phosphorylation of Viral and Host Proteins

The HCMV-encoded protein kinase pUL97 is a tegument protein, which is packaged into virions and is expressed with early–late kinetics [93,94,95,96]. This serine–threonine kinase possesses two nuclear localization signals (NLS) sequences in its N-terminus that mediate a predominantly nuclear localization in the HCMV-infected cells [97,98]. Dimers and oligomers are formed via the self-interaction domain (amino acids 231–280) of pUL97 [99]. Interestingly, the direct association of pUL97 with human cyclins has been demonstrated, suggesting a role in substrate recognition (cyclin B1) or pUL97 dimerization and oligomerization (cyclin T1 and H), respectively [86]. Interestingly, pUL97 undergoes massive autophosphorylation and shows a strong interaction with and phosphorylation of other viral tegument proteins, such as pUL83/pp65 and pUL69 [100]. Concerning the properties of protein interaction and substrate phosphorylation of pUL97, a number of viral and cellular proteins have been identified so far. The functionality of these substrates spans various regulatory aspects of viral replication, such as nuclear egress, intrinsic immunity, genome replication, and gene expression [101]. The spectrum of currently identified pUL97 protein substrates comprises viral and host proteins. The viral targets of pUL97 comprise the DNA polymerase cofactor pUL44 [102,103], the RNA transport factor pUL69 [87,104,105], the major tegument protein pUL83/pp65 [26,100,106], and the nuclear egress core protein heterodimer pUL50–pUL53 [35,107]. The host proteins shown to be phosphorylated by pUL97 are the cellular multiligand binding protein p32/gC1qR [108], the tumor suppressor and checkpoint protein Rb [109,110,111], its associated protein E2F3 [112], the Rb-related proteins p107 and p130 [113] and their interacting protein LIN52 [112], the nuclear lamins A/C [108,114,115], the RNA polymerase II [116,117,118], the translation factor EF-1δ [119,120], the FOXM1 transcription factor [112], the interferon-inducible protein IFI16 [121], and the host restriction factor SAMHD1 [122].

The specific sequence motif targeted by the pUL97 kinase activity in substrate proteins is still a matter of debate. Indeed, based on results from in vitro kinase assays, Baek et al. reported that the motif SxxxxK/R may be preferentially recognized by pUL97 [123]. This finding was later challenged by the analysis of phosphopeptide sequences identified by screening differentially abundant phosphorylation sites in human cells infected with either wild-type HCMV or kinase-inactive mutant HCMV (pUL97 K355Q) [25]. This study suggested LxSP and SxxK as putative pUL97-recognized motifs for substrate phosphorylation. So far, these motifs have been helpful to some extent in the characterization of pUL97-targeted substrates and binding domains. It should be noted, however, that due to the complexity of pUL97 interactions undergone with substrate proteins, cyclins, bridging factors, and multiprotein complexes [11,86,88,108], a safe mode of bioinformatic predictability has seemingly not been achieved so far.

Several of the pUL97-specific substrate proteins are also substrates of cellular cyclin-dependent kinase (CDK)–cyclin complexes, and thus may underlie a process of dual phosphorylation in HCMV-infected cells. For some of the substrate proteins, the specific target phosphosites of pUL97 have been identified and some of these phosphosites are even identical with those recognized by CDKs. In particular for several host proteins phosphorylated by pUL97, this dual mode of site-specfic phosphorylation has been identified, such as for lamins A and C (e.g., S22), retinoblastoma protein Rb (e.g., T356, T373, S608, S612, S780, S788, S795, S807, S811, T821, and T826), and SAMHD1 (e.g., T592) [111,112,113,114,115,122,124]. For other cellular proteins subject to pUL97-specific phosphorylation, such as the intrinsic immune factor IFI16 [121], less information was provided in terms of phosphosites.

Together with absent in melanoma *2* (AIM2), IFI16 belongs to the family of inflammasome-associated pathogen recognition receptors (PRRs), described as AIM2-like receptors (ALRs) [125]. We have previously shown that overexpressed IFI16 blocks HCMV replication by inhibiting the binding of Sp1 to the UL54 promoter [126]. To counteract this restriction activity, the tegument protein pUL83/pp65, in the early phase of infection, hijacks IFI16 to activate the major immediate–early promoter–enhancer (MIEP) expression, whereas later during infection, pUL83/pp65 interacts with IFI16 at the promoter of the UL54 gene and downregulates viral replication, as shown by using a HCMV mutant-lacking pUL83/pp65 expression [114]. Interestingly, at late replicative time points, HCMV is also able to promote the delocalization of IFI16 from the nucleus to the cytoplasm through pUL83/pp65, involving a pUL97-mediated mechanism [121,127]. Phospho-IFI16 is finally redirected to the viral assembly complex, where it is incorporated into newly formed viral particles [121]. In the context of the present study, we performed MS-based characterization of IFI16 phosphorylation sites, either depending on or independently of pUL97 activity. To this end, we used a transient transfection system in 293T cells to overexpress the required amounts of a previously characterized, recombinant version of IFI16 (IFI16ΔD-V5) [126], either alone or under co-expression of a catalytically active form of pUL97 (pUL97-Flag) [88]. Using total cellular lysates, protein complexes could be harvested upon IFI16-specific co-immunoprecipitation by a V5-specific monoclonal antibody (for co-immunoprecipitation procedures see [126]). Two specificity control samples were analyzed in parallel for monitoring the pUL97 activity, i.e., on the one hand an inactive mutant of pUL97 used for IFI16 co-expression (K355M), while on the other hand treatment with the pharmacological pUL97 inhibitor maribavir (5 µM MBV). Two independent experiments were performed and analyzed. As an important result, our MS-based phosphoproteomic analysis revealed eleven phosphosites in IFI16 (i.e., S106, S153, S160, S211, S418, S445, S562, S568, S575, S588, and S744) identified with a localization probability above 75% (Table 1). Six of them (i.e., S211, S418, S445, S562, S588, and S744) had not been previously shown to be phosphorylated (https://www.nextprot.org/entry/NX_Q16666/proteomics, [128]). The abundances of eight IFI16 phosphosites were reliably extracted using Proline [37] in the two independent replicates. Their comparison in the pUL97-positive and control samples, after normalization using unphosphorylated IFI16 peptides [129], revealed that five could be assigned to pUL97 activity, namely S106, S211, S445, S588, and S744, while the three others seem to be phosphorylated independently of pUL97, i.e., S153, S568, and S575 (Figure 5). Importantly, it was impossible to discriminate the quantification of S568 and S575, since they are carried by isobaric and co-eluting peptides, even if the identified peptides were not carrying both modifications together, making it possible that one could be targeted by pUL97. In conclusion, this analysis identified specific phosphorylated serine and threonine residues in IFI16, and suggested that viral pUL97 and host kinases are responsible for the modification of these sites. Future experimentation is needed to clarify the functional roles played by each of these pUL97-dependent IFI16 phosphosites, either to counteract the intrinsic immunity response or to promote the nucleocytoplasmic translocation of IFI16 in HCMV-infected cells.

## 6. Conclusions and Perspectives

The assembly of the complex HCMV virion is orchestrated by multiple protein interactions. The molecular mechanisms that drive these processes are widely unknown. Their elucidation using modern technologies of proteome analysis is essential for the understanding of herpesvirus morphogenesis. The comprehensive knowledge gathered so far and reviewed in this communication provides a basis for a detailed analysis of the relevance of individual protein interactions and their regulation by modifications such as phosphorylation. The information drawn from these data sets will furthermore be instrumental for the development of antiviral compounds that specifically target morphogenesis.

## Figures and Tables

**Figure 1 microorganisms-08-00820-f001:**
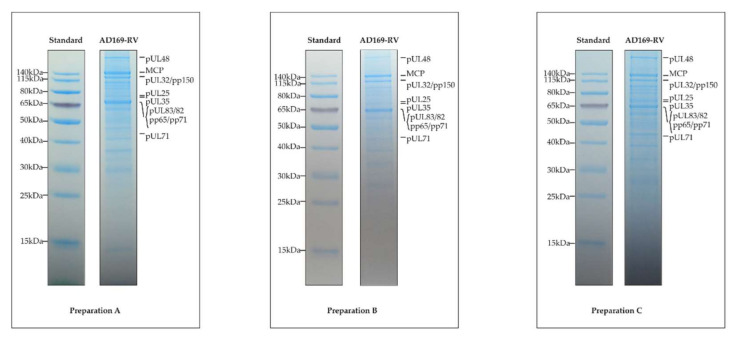
Protein patterns of the three replicates of purified human cytomegalovirus (HCMV) virions. Extracted proteins of the three different preparations were submitted to SDS-PAGE separation and stained with Instant Blue dye. The proteins depicted to the right in each panel were tentatively assigned according to the protein molecular weight standard, shown to the left.

**Figure 2 microorganisms-08-00820-f002:**
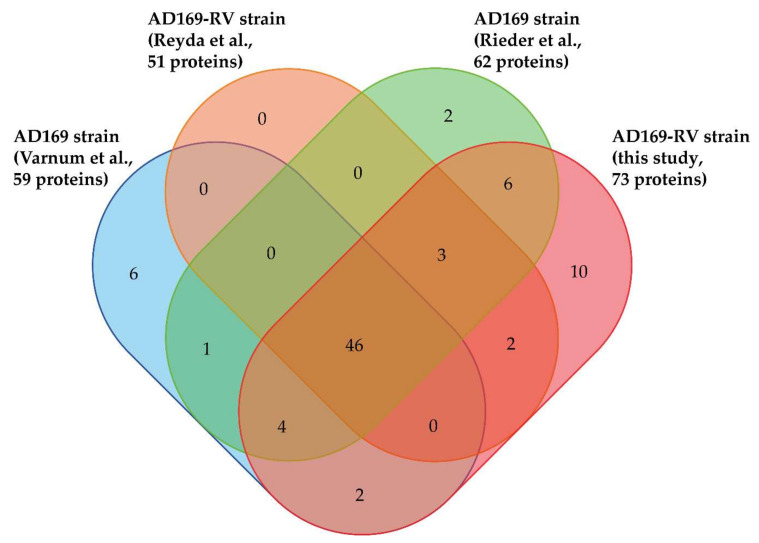
Venn diagram displaying overlap of viral proteins identified in HCMV virions in the present and previous studies [18,19,23] (http://bioinformatics.psb.ugent.be/webtools/Venn/).

**Figure 3 microorganisms-08-00820-f003:**
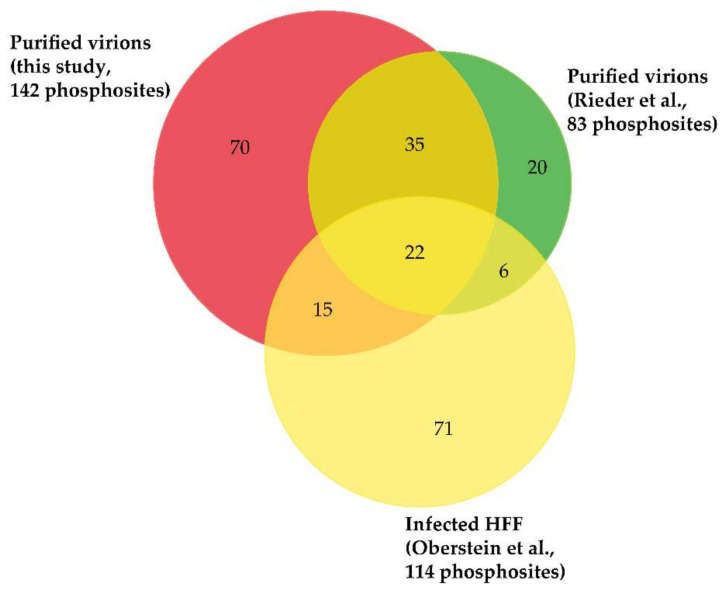
Venn diagram displaying overlap of phosphosites identified on viral proteins in HCMV virions (Rieder et al. [23] and this study) and in infected cells (Oberstein et al. [25]). Importantly, only phosphosites identified with localization probabilities above 75% were taken into account.

**Figure 4 microorganisms-08-00820-f004:**
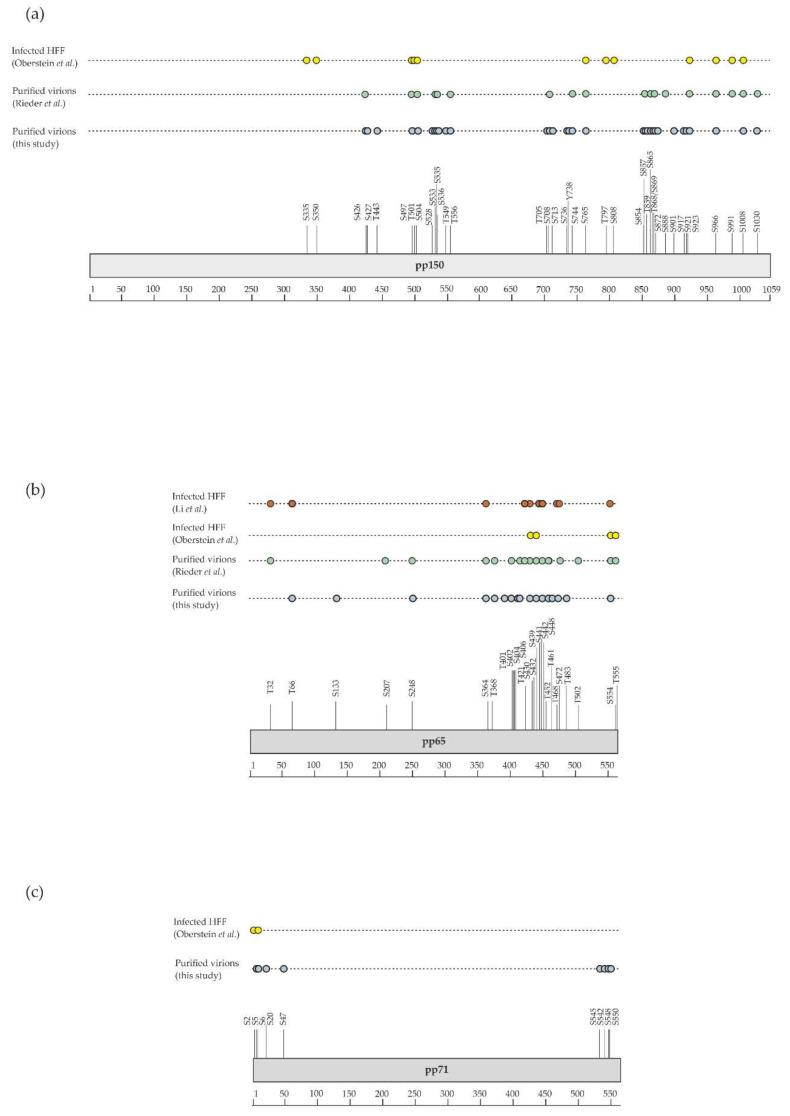
Graphical display of phosphosites identified in this study and by previous investigations (details are given in Appendix A). Shown are phosphosites found on (**a**) pUL32/pp150, (**b**) pUL83/pp65, (**c**) pUL82/pp71, (**d**) pUL25, (**e**) pUL97, and (**f**) pUL69 in virions (this study and Rieder et al. [23]) and in infected cells [25,83,84,85,86,87] (HFF, human foreskin fibroblasts).

**Figure 5 microorganisms-08-00820-f005:**
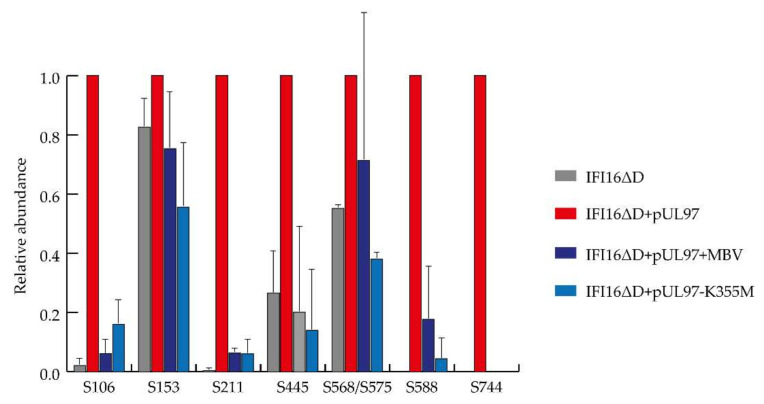
Dependence of IFI16 phosphosites on pUL97 kinase activity determined by mass spectrometry (MS)-based phosphoproteomic analysis. Extracted abundances of each phosphosite were normalized on the IFI16ΔD + pUL97 sample and averaged between the two independent replicates. Error bars represent standard deviation.

**Table 1 microorganisms-08-00820-t001:** Identification of IFI16 phosphosites by MS-based phosphoproteomic analysis. Only sites identified with a localization probability above 75% were considered.

Identified IFI16 Phosphosites	Best Localization Probability
S106	96%
S153	100%
S160	87%
S211	84%
S418	85%
S445	89%
S562	96%
S568	100%
S575	100%
S588	100%
S744	96%

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
