# Peer review of "Mass Spectrometry-Based Characterization of the Virion Proteome, Phosphoproteome, and Associated Kinase Activity of Human Cytomegalovirus"

_microorganisms, 2020, doi:10.3390/microorganisms8060820_

Round 1

Reviewer 1 Report

General comments.

In the paper “Mass spectrometry-based characterization of the virion proteome, phosphoproteome and associated kinase activity of human cytomegalovirus.” Authors reported a review concernin a comprehensive map of the HCMV virion proteome, including a refined view on the virion phosphoproteome, based on previous publications supplemented by new results. Thus, a novel dataset of viral and cellular proteins contained in HCMV virions has been generated, providing. Finally, authors present the current knowledge on the activity of pUL97, the HCMV-encoded and virion-associated kinase, and deliver original results on the phosphorylation of the host protein IFI16.

Paper is good write. I suggest minor revisions.

Specific comments.

Page 3 line 55

“For virion purification 1.8x106 primary HFF cells were grown in 175-cm2]”

Please, report cm2  

Page 13 line 432

Figure 5, pleas add error bars

General comments.

In the paper “Mass spectrometry-based characterization of the virion proteome, phosphoproteome and associated kinase activity of human cytomegalovirus.” Authors reported a review concernin a comprehensive map of the HCMV virion proteome, including a refined view on the virion phosphoproteome, based on previous publications supplemented by new results. Thus, a novel dataset of viral and cellular proteins contained in HCMV virions has been generated, providing. Finally, authors present the current knowledge on the activity of pUL97, the HCMV-encoded and virion-associated kinase, and deliver original results on the phosphorylation of the host protein IFI16.

Paper is good write. I suggest minor revisions.

Specific comments.

Page 3 line 55

“For virion purification 1.8x106 primary HFF cells were grown in 175-cm2]”

Please, report cm2  

Page 13 line 432

Figure 5, pleas add error bars

Author Response

We are grateful to Reviewer 1 for her/his very positive comments supporting the publication of our manuscript, as well as for suggestions to improve it.

Specific comments :

1) Please, report cm2  

This has been corrected in the revised manuscript.

2) Figure 5, pleas add error bars

In the submitted version, we did not plot error bars because the presented results were coming from a single experiment. In order to strengthen the conclusions in the revised version of our manuscript, we combined results from two independent experiments. The corresponding section was modified accordingly and error bars have been added.

Reviewer 2 Report

This review article by Couté et al. summarizes the current knowledge of the virion proteome and phosphoproteome. It also includes additional unpublished data and compares it with the previously published results. This is a very useful review for the field as is allows the definition of a „core“ or „consensus“ virion proteome and a set of viral and cellular proteins identified only in a few assays, indicating that the proteins involved are either less abundant in the virion or less reliable components of viral particles.

The paper is well structured, contains very nice illustrations, and is easy to read. Nevertheless, I have a few points for the authors‘ consideration.

(1) This review and the data sets compared are all based on the laboratory strain AD169. However, we all know that this strain is somewhat atypical due to its extensive laboratory adaptation. Have the virion proteomes of other strains been analyzed, and if so, how do the proteomes compare to the ones of strain AD169?

(2) The new data on the phosphorylation of IFI16 does not really fit with the rest of the story. I strongly suggest to delete this dataset entirely or tone it down significantly (remove reference to this in the abstract). Lines 424-426. What do we learn from knowing these phosphorylation sites? The reader needs to know why is this important and why it should be included in a review on the virion proteome.

(3) Line 140. Please comment on the completeness of the Uniprot database. The HCMV genome has been annotated and re-annotated several times. Please comment on how accurately the Uniprot database reflects the current knowledge of the true HCMV proteome and which proteins might have been missed because of the incompleteness of the Uniprot database.

(4) Line 98. HB15 is not a proper strain name. This is a BACmid clone of strain AD169. To my knowledge, the proper reference for this strain is PMID 10933677, where this BACmid was first described.

 (5) Line 162. Alloherpesviridae is the name of a virus family. Thus the sentence should read: that of Ictalurid herpesvirus 1 (also known 162 as Channel Catfish Virus), a member of the  Alloherpesviridae,

Author Response

We are grateful to Reviewer 2 for her/his very positive comments supporting the publication of our manuscript, as well as for suggestions to improve it.

(1) This review and the data sets compared are all based on the laboratory strain AD169. However, we all know that this strain is somewhat atypical due to its extensive laboratory adaptation. Have the virion proteomes of other strains been analyzed, and if so, how do the proteomes compare to the ones of strain AD169?

As emphasized by Reviewer 2, we forgot to mention in the previous version of our manuscript that we already addressed this issue in a previous paper (Büscher et al. PMID:25732096). In this work, we showed that the protein composition of HCMV virions is conserved between various HCMV strains, notably using a laboratory strain, two isolates that had been passaged many times on fibroblasts, thereby losing the expression of the pentameric complex and the TB40-BAC4 clone of the TB40/E strain that still expresses the pentamer. We modified the revised manuscript accordingly (end of penultimate paragraph of chapter 3) and added the corresponding reference.

(2) The new data on the phosphorylation of IFI16 does not really fit with the rest of the story. I strongly suggest to delete this dataset entirely or tone it down significantly (remove reference to this in the abstract). Lines 424-426. What do we learn from knowing these phosphorylation sites? The reader needs to know why is this important and why it should be included in a review on the virion proteome.

In the fifth chapter of our review, we present the current knowledge on the activity of pUL97, the HCMV-encoded and virion-associated kinase. Among its presented cellular targets, IFI16 is of particular interest to us since the pUL97-dependent phosphosites have never been described, and because IFI16 has been shown as a virion-associated protein, which is delivered into cells upon infection. We therefore strongly thinks that it is interesting to present these novel data in this manuscript. Furthermore, so far, there has been very little data published on the phosphorylation of intrinsic immune restriction factors by pUL97. It is the first time that distinct phosphosites in IFI16 are presented in this context and this regulatory phenomenon (possibly a way of down-modulating that activity of IFI16, or promoting its pUL97-dependent nucleocytoplasmic translocation in HCMV-infected cells, as previously published by our group) fits very well into the current concept that HCMV developed an immediate counteracting response towards the intrinsic immunity. We added a specific sentence at the end of this chapter.

However, to take into account the remark of Reviewer 2, we agreed to tone down the findings to some extent as we removed the reference to IFI16 in the abstract of the revised version of our manuscript.

(3) Line 140. Please comment on the completeness of the Uniprot database. The HCMV genome has been annotated and re-annotated several times. Please comment on how accurately the Uniprot database reflects the current knowledge of the true HCMV proteome and which proteins might have been missed because of the incompleteness of the Uniprot database.

This is indeed a very important point reported by Reviewer 2. We decided to specifically add a sentence on that particular point in the third paragraph of chapter 3, notably citing the Stern-Ginossar et al. work (PMID: 23180859) that highlighted the unexpected complexity of the HCMV coding capacity and revealed novel potential protein products. To our knowledge, the Uniprot database content reflects the current knowledge about theoretical HCMV proteome. As an example, it contains the potential protein products provided in the Stern-Ginossar et al. publication. However, since database contents are constantly evolving according to advances in biological knowledge, it is important that MS-based proteomic data can be reprocessed in the future. This is why we decided to deposit our raw and processed data in the ProteomeXchange repository (dataset identifier PXD012921) to make them publically available upon the acceptance for publication of the manuscript. The text was modified accordingly in the chapter 2.2.

(4) Line 98. HB15 is not a proper strain name. This is a BACmid clone of strain AD169. To my knowledge, the proper reference for this strain is PMID 10933677, where this BACmid was first described.

As suggested, we have changed the denomination of the viral strain to AD169-RV in the text and in Figures 1 and 2. The reference was modified accordingly.

 (5) Line 162. Alloherpesviridae is the name of a virus family. Thus the sentence should read: that of Ictalurid herpesvirus 1 (also known 162 as Channel Catfish Virus), a member of the  Alloherpesviridae,

We modified the text as suggested.

Reviewer 3 Report

In this manuscript Coute et al. provide a new data set for HCMV virion proteome and phosphoproteome, including proteins of viral and host origin, and they put together their data with previously published data sets to review what we know about HCMV virion assembly and proteome. Methods are explained in great detail and the paper is very well written and organized. To my knowledge, this is the first review on HCMV virion proteomics. In my opinion, this is a very good review that will be of interest for scientists working on many different aspects of HCMV biology.

Author Response

We are grateful to Reviewer 3 for her/his very positive comments supporting the publication of our manuscript.

Reviewer 4 Report

This paper is very well written, clear and carefully prepared. The research work is highly technical and well performed. The work is examining the protein composition of the HCMV particles and provides important scientific evidence that can be used for further studies of new pharmaceutical targets. The study is partly supporting earlier evidence that has been referenced and is also bringing in some new data and aspects.

There are some major comments and questions to the authors: 

1) The study results are referring to three datasets investigating virus particles of the HCMV laboratory AD169 strain. The earlier data that has been referenced also investigated particles of the same viral strain. It is therefore understandable that the authors chose to use the same viral strain to be able to relate the current to earlier results. However, it is well documented that the laboratory strains differ from the clinical strains (which have also been discussed in this paper) influencing their pathogenicity. Since HCMV is known to be highly variable virus, also the different clinical strains are known to differ in infectivity and pathogenicity. Why didn´t the authors investigate the viral particles from any clinical strains in addition to the AD169 strain? Wouldn't that have been more clinically relevant since the important proteins of the pentameric complex are deleted/mutated in the AD169 strain and these proteins are the key proteins for the viral entry? Wouldn't the use of a clinical strain have added important, clinically relevant, new information?

2) Some conclusions of this paper state that the protein content of viral particles are highly conserved based on this and the earlier studies. However, can this conclusion be drawn since these studies only investigated one laboratory viral strain (AD169)?

3) The finding of host proteins in the viral particles is reported in this study as well as earlier studies. The number of the host proteins is reported to vary between 63-1000 proteins depending on the used inclusion criteria. The role and locations of the host proteins remains to be elucidated. The authors report abundant presence of Annexin A2 known to be present on the surface of the virions and being important for the infectivity of the viral particles. The host proteins may play several functions, but how about the immunological aspect of these proteins for the virus particles? Reasonably, the virus may use the host proteins, especially glycoproteins, on its surrounding lipid envelope and therefore make it look like material from its host. Is there any supporting evidence for this among these results? Could this be named or discussed? 

4) The study reports a number of new potential phosphosites on the key viral and host proteins in the virus particles. Do the authors believe that some of these would be essential for the viral entrance or infectivity and since located inside the outer viral membrane, how could they be used as potential new drug targets?

Author Response

We are grateful to Reviewer 4 for her/his very positive comments supporting the publication of our manuscript, as well as for suggestions to improve it.

1) The study results are referring to three datasets investigating virus particles of the HCMV laboratory AD169 strain. The earlier data that has been referenced also investigated particles of the same viral strain. It is therefore understandable that the authors chose to use the same viral strain to be able to relate the current to earlier results. However, it is well documented that the laboratory strains differ from the clinical strains (which have also been discussed in this paper) influencing their pathogenicity. Since HCMV is known to be highly variable virus, also the different clinical strains are known to differ in infectivity and pathogenicity. Why didn´t the authors investigate the viral particles from any clinical strains in addition to the AD169 strain? Wouldn't that have been more clinically relevant since the important proteins of the pentameric complex are deleted/mutated in the AD169 strain and these proteins are the key proteins for the viral entry? Wouldn't the use of a clinical strain have added important, clinically relevant, new information?

As emphasized by Reviewer 4, we forgot to mention in the previous version of our manuscript that we already addressed this issue in a previous paper (Büscher et al. PMID:25732096). In this work, we showed that the protein composition of virions is conserved between several HCMV strains, notably a laboratory strain, two isolates that had been passaged many times on fibroblasts, thereby losing the expression of the pentameric complex and the TB40-BAC4 clone of the TB40/E strain that still expresses the pentamer. We modified the revised manuscript accordingly (end of penultimate paragraph of chapter 3) and added the corresponding reference.

2) Some conclusions of this paper state that the protein content of viral particles are highly conserved based on this and the earlier studies. However, can this conclusion be drawn since these studies only investigated one laboratory viral strain (AD169)?

See response to question 1.

3) The finding of host proteins in the viral particles is reported in this study as well as earlier studies. The number of the host proteins is reported to vary between 63-1000 proteins depending on the used inclusion criteria. The role and locations of the host proteins remains to be elucidated. The authors report abundant presence of Annexin A2 known to be present on the surface of the virions and being important for the infectivity of the viral particles. The host proteins may play several functions, but how about the immunological aspect of these proteins for the virus particles? Reasonably, the virus may use the host proteins, especially glycoproteins, on its surrounding lipid envelope and therefore make it look like material from its host. Is there any supporting evidence for this among these results? Could this be named or discussed? 

We agree that this is indeed an interesting point. We therefore discussed further this topic in the last paragraph of chapter 3. However, since no clear evidence based on data is available at this stage, a detailed statement is not possible, but could be obtained by further investigations in the future, as this review attempts to stimulate.

4) The study reports a number of new potential phosphosites on the key viral and host proteins in the virus particles. Do the authors believe that some of these would be essential for the viral entrance or infectivity and since located inside the outer viral membrane, how could they be used as potential new drug targets?

We agree that this is indeed an interesting point. We therefore discussed further this topic in a novel paragraph at the end of chapter 4. However, as for the previous point, since no clear evidence based on data is available at this stage, a detailed statement is not possible, but could be obtained by further investigations in the future, as this review attempts to stimulate.